# Mechanism of N-Acetyl-D-alloisoleucine in Controlling Strawberry Black Root Rot

**DOI:** 10.3390/plants14050829

**Published:** 2025-03-06

**Authors:** Jialu Xu, Jianxiu Hao, Mingmin Zhao, Xiaoyu Zhang, Ruixiang Niu, Yiran Li, Zhen Wang, Shuo Zhang, Sumei Zhao, Siran Li, Hongyou Zhou

**Affiliations:** 1College of Horticulture and Plant Protection, Inner Mongolia Agricultural University, Hohhot 010020, China; xujialu@emails.imau.edu.cn (J.X.); haojianxiu0402@imau.edu.cn (J.H.); mingminzh@163.com (M.Z.); zxy2000@126.com (X.Z.); nrx000123@emails.imau.edu.cn (R.N.); zhangs047@163.com (S.Z.); 2Xilin Gol League Agricultural and Animal Husbandry Technology Extension Center of Inner Mongolia, Xilinhot 026000, China; yiran1701@163.com; 3Ulanqab Agricultural and Forestry Science Research Institute of Inner Mongolia, Ulanqab 012209, China; wangzhenwlcb@163.com; 4Plant Protection and Quarantine Center of Inner Mongolia Autonomous Region, Hohhot 010010, China; smzhaohh@126.com; 5Xingan League Meteorological Bureau of Inner Mongolia, Ulanhot 137400, China; 15148940381@163.com

**Keywords:** strawberry black root rot, *F. asiaticum*, N-acetyl-D-alloisoleucine, inhibition effect, resistance

## Abstract

China is the largest strawberry producer in the world. Strawberry black root rot is a novel disease that occurs in Hohhot, Inner Mongolia. In the present study, the inhibitory effects of *Bacillus subtilis* S-16 and its fermented form on strawberry black root rot caused by *Fusarium asiaticum* were tested. The inhibition rates were 56.31% and 65.95%, respectively. Furthermore, the metabolic substances were analysed using LC-MS/MS. A total of 68 substances were identified, including 18 amino acids, 7 of which have been reported to have pro-growth and antibacterial functions. Among these seven amino acids, N-acetyl-D-alloisoleucine (NAD) had the strongest inhibitory effect on *F. asiaticum*. In addition, NAD caused the mycelia of *F. asiaticum* to appear shrivelled and deformed under electron microscopy. Furthermore, the effect of NAD on *F. asiaticum* was tested. The results indicate that NAD had a better prevention effect when used with hymexazol. Finally, the fungal biomass of *F. asiaticum* in strawberry roots was measured at different times using two treatment methods: treating plant roots with NAD and a spore suspension of *F. asiaticum* concurrently and with *F. asiaticum* alone. The colonisation response of *F. asiaticum* in terms of the target gene EF-1α when treated with *F. asiaticum* alone at 72 hpi was significantly higher than that when treated with NAD and a spore suspension of *F. asiaticum*. The relative expression levels of defence-related genes in strawberry roots treated with NAD at 72 hpi were determined. The genes NPR1 and PDF1 were markedly upregulated compared with other genes, suggesting that the expression of genes related to disease resistance was activated by NAD, resulting in disease resistance in strawberries. Our results provide theoretical support for the biological control of strawberry black root rot.

## 1. Introduction

Strawberry (*Fragaria ananassa* Duch.), a perennial herb of the strawberry genus, belonging to the rose family, is an important berry crop native to South America, with a long history of cultivation. Strawberries are rich in various nutrients, such as vitamins C, A, and E, which have excellent nutritional and health benefits [1,2,3]. Strawberry production was valued at USD 14 billion in 2020, as reported by the Food and Agriculture Organization of the United Nations (FAO) in 2021. China is the largest strawberry producer worldwide, accounting for USD 5 billion, over three times the value of the second-largest producer [4]. In recent years, with the development of agricultural facilities in Inner Mongolia, strawberries have attracted much attention as a crop with high economic benefits. Especially in the city of Hohhot, the planting area has been increasing annually, effectively promoting the development of industries such as sightseeing, picking, and rural tourism. The strawberry industry has played a positive role in increasing farmers’ incomes and improving agricultural efficiency, achieving the dual promotion of economic and social benefits [5]. However, with the annual increase in strawberry planting area, strawberries are susceptible to various diseases during the cultivation process, such as grey mould, wilt disease, and powdery mildew [6]; in particular, since 2019, a very serious unknown disease has occurred in the Hohhot area. This disease can occur throughout the growth period, but especially during the seedling stage. Infected plants generally have blackened roots, dark brown leaves, poor development, and later wither or even die within 15 days. The plant mortality rate exceeds 90%, causing significant economic losses for farmers. After its identification, it was found to be caused by *Fusarium asiaticum*, which is also known as “black rot” disease [5].

The use of numerous chemical pesticides for the prevention and control of plant diseases has caused serious environmental pollution and threatens food safety and human health. Currently, the exploration of environmentally friendly biological control methods is a popular research topic [7]. Various biological agents, such as bacteria, fungi, and their metabolites, have been used to control plant diseases in tomatoes, cucumbers, watermelons, and other crops [8,9,10]. The use of secondary metabolites is a biological strategy for the management of plant diseases. Secondary metabolites are a large group of substances with low molecular weights, low polarities, low boiling points, and high vapour pressure [11]. They often belong to different chemical classes, such as alcohols, benzenoids, aldehydes, alkenes, carbonyl compounds, amino acids, peptides, and their derivatives [12,13]. Many of these metabolic compounds (MCs) have biotechnological applications in agriculture and industry [14]. The use of MCs derived from various *Bacillus* species as fungicides has also increased. The cyclic peptide RN from *Bacillus subtilis* RN8 damages the outer membrane of *Candida albicans* cells, destroys the overall structure of the cell membrane, leads to the leakage of intracellular substances, and significantly affects the main metabolic pathways within the cell, thus playing an antibacterial role [15]. Active antimicrobial peptides, such as subtilin and subtilosin A isolated from *B. subtilis* BS-6, include small-molecule antimicrobial peptides synthesised through non-ribosomal synthesis and large-molecule protein bacteriocins synthesised through ribosome synthesis [16]. Yanqin et al. isolated and screened *B. subtilis* from a natural sea area and obtained crude extracts of active antibacterial substances from the fermentation broth. After treatment with papain and protease K, antibacterial activity disappeared, suggesting that the substance was proteinaceous [17]. It can be seen that *B. subtilis*, as a functional antibacterial microorganism, can produce a variety of antibacterial substances, such as peptide substances and non-peptide compounds [18]. However, there are few reports on the use of amino acids derived from the metabolic fluids of *B*. *subtilis* for plant disease prevention and control.

In our previous studies, the volatile compound 2-methylbenzothiazole from *B. subtilis* S-16 inhibited the mycelial growth of *Sclerotinia sclerotiorum* and *Botrytis cinerea*. The combination ratio of S-16 and PT-29 of 2:1 exhibited the best control effects against potato *Verticillium* wilt, and there were more highly abundant amino acids in the 2:1 S-16 and PT-29 combination compared to single-cultured S-16 and PT-29 which were correlated with amino acids in S-16 but not in PT-29 through liquid LC-MS and metabolomics analysis [19,20]. In the present study, *B. subtilis* S-16 and its fermentation broth were subjected to bacteriostatic tests to determine their ability to control *F. asiaticum* CM-1. Secondary metabolites were also screened using LC-MS/MS metabolomic analysis. Then, we verified the preventive effect displayed by the target substance N-acetyl-D-alloisoleucine against *F. asiaticum* and strawberry “black rot” disease in experiments. Our findings provide new ideas for the biological control of strawberry “black rot” disease.

## 2. Results

### 2.1. Inhibition of F. asiaticum CM-1 by B. subtilis S-16 and Its Fermentation Broth

*B. subtilis* S-16 and its fermentation solution both had strong inhibitory effects on *F. asiaticum* CM-1. The inhibition rates were 56.31% and 65.95%, respectively (Figure 1A,B). These results show that *B. subtilis* S-16 or its fermentation solution can be used to control strawberry black root rot. Some secondary metabolites from the fermentation solution of S-16 may have inhibitory effects on *F. asiaticum* CM-1.

### 2.2. Analysis of MCs Produced by B. subtilis S-16

According to Result 1, MCs were analysed using LC-MS/MS. A total of 68 metabolites, including amino acids, carbohydrates, and fats, were annotated. Among them, amino acids were the most abundant, with a total of 18 types; 3 types were significantly upregulated, while 15 types were significantly downregulated (Appendix A). Of these 18 amino acids, 7 were reported to be associated with the inhibition of pathogenic bacteria [21,22,23,24,25,26,27] (Table 1).

### 2.3. Effect of N-Acetyl-D-alloisoleucine on the Inhibition of F. asiaticum CM-1 Growth

The standard products of seven amino acids with antibacterial functions and their inhibitory effects on CM-1 were determined using the Oxford cup test. The results show that L-aspartic acid, ornithine, L-glutamic acid, and N-acetyl-D-alloisoleucine had significant inhibitory effects on CM-1, with N-acetyl-D-alloisoleucine having the best inhibitory effect, with an inhibitory rate of up to 69.45% (Figure 2A,B). Therefore, we used NAD as the target and further observed the mycelial morphology under NAD treatment via electron microscopy. The results show that the mycelia of CM-1 appeared to be shrivelled and deformed after NAD treatment. This indicates that NAD may inhibit the mycelial formation of CM-1 (Figure 2C).

### 2.4. The Preventive Effect of N-Acetyl-D-alloisoleucine on Strawberry Black Root Rot

Four groups were established for the pot experiments. The treatment groups were treated simultaneously with NAD and an *F. asiaticum* spore suspension at a concentration of 1 × 10^6^ spores/mL. Another treatment involved simultaneous treatment with metalaxyl, which had good effects on plant root rot [28], and *F. asiaticum*. The positive control comprised strawberry roots treated with only *F. asiaticum*, with a spore concentration of 1 × 10^6^ spores/mL, and the blank control was treated with sterile water. After 10 days, the disease index of black root rot was established, and the control efficiency was calculated. The results show that NAD and metalaxyl had a better preventive effect than the positive control, and plants in the blank control group did not show any disease symptoms (Figure 3A–D). The NAD control effect was 69.75%, which was similar to that observed in metalaxyl (Figure 3E). This demonstrates that NAD has excellent biocontrol potential against strawberry black rot.

### 2.5. Determination of Colonisation Level of F. asiaticum in Strawberry Roots

The fungal biomass of *F. asiaticum* CM-1 in strawberry roots was measured at different time points (0 hpi and 72 hpi) using two treatment methods: treating plant roots with NAD and a spore suspension of CM-1 and with CM-1 alone. Additionally, the results show significant differences in the amount of root colonisation at 0 hpi and 72 hpi under the two treatment methods. The colonisation response of *F. asiaticum* in terms of the target gene EF-1α when treated with CM-1 alone at 72 h was significantly higher than that when treated with NAD and CM-1, which confirmed that NAD can inhibit the colonisation of CM-1 in strawberry roots (Figure 4).

### 2.6. Effect of N-Acetyl-D-alloisoleucine Treatment on the Relative Expression of Related Defence Genes in Strawberry Roots

In this study, we determined the relative expression levels of defence-related genes in strawberry roots. The defence-related genes PR1, PR5, NPR1, PDF1, and PAL were identified at 72 hpi after cells were treated with N-acetyl-D-alloisoleucine. The genes NPR1 and PDF1 were markedly upregulated compared with other genes, suggesting that the expression of genes related to disease resistance was activated by N-acetyl-D-alloisoleucine, resulting in improved disease resistance in strawberries (Figure 5).

## 3. Discussion

Strawberry black rot disease, first reported in Inner Mongolia, China, can occur throughout the entire growth period, and its onset is rapid. Within 10–15 days, the plants reach the most severe stage, causing a yield loss of up to 90% [5]. In our previous study, the inhibition rate of a *B. subtilis* ZWZ-19 and *T. asperellum* PT29 fermentation broth mixed at a ratio of 1:1 was 61.25% against potato wilt disease caused by *F. oxysporum* [12]. In the present study, we tested the inhibitory effects of *B. subtilis* S-16 and its fermentation solution on *F. asiaticum*, which causes strawberry black root rot. In addition, we analysed the metabolites of S-16, selected seven amino acids with reported antibacterial effects, and determined their inhibitory effects on *F. asiaticum* using the Oxford cup experiment. Based on the test results, we targeted N-acetyl-D-alloisoleucine to determine its preventive and controlling effects on strawberry black rot. Our results provide theoretical support for the biological control of strawberry root rot and the green and sustainable development of the strawberry industry.

*B. subtilis* has been designated as an excellent choice for the biological control of pathogenic microorganisms [29]. Several studies have demonstrated that *Bacillus* and its fermentation serve as positive strategies against plant diseases such as cotton *Verticillium* wilt, cucumber *Fusarium* wilt, and oilseed rape Sclerotinia stems [30,31,32]. In this study, the inhibitory effects of *B. subtilis* S-16 and its fermented form on strawberry black rot caused by *F. asiaticum* CM-1 were tested (Figure 1). The experimental results support the above report. The *Bacillus* genus is known for its production of multiple antimicrobial peptides (AMPs) composed of amino acids, positioning it as a promising candidate in the exploration of new inhibitory compounds [33]. We analysed the secondary metabolites in the fermentation broth of S-16 using the LC-MS/MS method and identified 68 types of substances, among which amino acids were the most abundant, with a total of 18 types; 3 types were significantly upregulated, while 15 types were significantly downregulated (Appendix A). Seven of these compounds were reported to inhibit pathogenic bacteria (Table 1). However, other studies have focused on enhancing the effects of *Bacillus* species on plant growth and disease control by optimising the fermentation solution [34,35]. This was the difference between our studies. Amino acids and their derivatives are not only the basic substances underlying microbial activities but also play a key role in coping with adversity and stress. Studying the functions of amino acids in microorganisms is important for understanding their physiological mechanisms, developing microbial resources, and addressing environmental issues. Amino acids can act as signalling molecules that regulate gene expression and metabolic pathways in microorganisms, exerting their related functions [36]. Seven types of amino acids with antibacterial functions were tested with regard to their inhibitory effects on CM-1, and NAD was found to have the best inhibitory effect, with an inhibitory rate of up to 69.45% (Figure 2A,B). Therefore, the morphology of the mycelia under N-acetyl-D-alloisoleucine treatment was observed using electron microscopy. The results indicate that NAD inhibited the mycelium formation of CM-1 (Figure 2C). This is the first report on the use of NAD in plant disease control, although its role in promoting maize growth has been reported previously [23].

In addition, the fungal biomass of the isolate *F. asiaticum* CM-1 in strawberry roots was measured at different times using two treatment methods: treating plant roots with NAD and a spore suspension of CM-1 and with CM-1 alone. In addition, the relative expression levels of defence-related genes in strawberry roots were tested at 72 hpi. The colonisation response of *F. asiaticum* in terms of the target gene EF-1α when treated with CM-1 alone at 72 h was significantly higher than that when treated with NAD and CM-1, which confirmed that NAD can inhibit the colonisation of CM-1 in strawberry roots (Figure 4). In addition, the defence-related genes PR1, PR5, NPR1, PDF1, and PAL were determined at 72 hpi after treatment with NAD. The genes NPR1 and PDF1 were markedly upregulated compared with other genes, suggesting that the expression of genes related to disease resistance was activated by NAD, resulting in improved disease resistance in strawberries (Figure 5). This finding is similar to that of our previous study, in which 6-PP not only inhibited the growth of *F*. *oxysporum* but also activated the relative expression levels of genes related to the disease course in tomatoes [37].

In summary, N-acetyl-D-alloisoleucine can not only directly inhibit the growth of *F. asiaticum* and control strawberry black root rot by inducing a high expression of the strawberry disease resistance genes NPR1 and PDF1 (Figure 6), but it can also serve as an important biocontrol substance to prevent and control strawberry black root rot and provide a theoretical basis for the green and sustainable development of the strawberry industry.

## 4. Materials and Methods

### 4.1. Pathogens, Biocontrol Strain, and Plants

The pathogen and biocontrol strains used in this study were obtained from the Fungus Preservation Collection of the Inner Mongolia Agricultural University, China. *F. asiaticum* CM-1, which causes strawberry “black rot” disease, was cultured on Petri dishes containing autoclaved potato dextrose agar (PDA) at 25 °C in a constant-temperature incubator for 5 days. The biocontrol strain, *B. subtilis* S-16, was also cultured on Petri dishes containing autoclaved Luria–Bertani (LB) medium at 28 °C in a constant-temperature incubator for 48 h. Strawberry plants of the variety “Hongyan” were provided by Xuefeng Yang from the Inner Mongolia Autonomous Region Institute of Biotechnology.

### 4.2. LC-MS/MS Analysis of the Fermentation Broth of B. subtilis S-16

*B. subtilis* S-16 was cultured on LB medium for two days, and a single colony was selected for further incubation in 100 mL of LB broth in a shaking incubator at 180 rpm and 28 °C for 24 h. Then, the OD_600_ value of the S-16 fermentation broth was measured using an ultraviolet spectrophotometer HD-UV90 (Holder electronic Technology Co., Ltd., Jinan, China). An OD_600_ value of 1.0 could ensure that S-16 was in the optimal logarithmic growth phase, which was what was needed. Then, the fermentation solution was centrifuged at 12,000 rpm at 4 °C for 15 min to obtain the supernatant [20]. The supernatant was filtered through a sterilised Millipore filter with a pore size of 0.22 μm and sent to Beijing Novogene Technology Co., Ltd., Beijing, China, for metabolomic testing.

We made slight modifications to the method proposed by Hu et al. (2023) [38]. All samples were thawed at 4 °C and mixed evenly. We placed 1 mL of the sample in a 1.5 mL centrifuge tube and centrifuge it at 4500 rpm/min for 10 min. Then, 200 μL of supernatant was added to 400 μL of the extract solution (methanol/acetonitrile = 3:1) and left to stand for 1 h at 4 °C following vortex oscillation. Next, the supernatant was centrifuged at 12,000 rpm/min and 4 °C for 15 min, and then an equal volume of a 50% methanol aqueous solution was added. Subsequently, 100 μL of the solution containing 50% methanol and each sample was centrifuged for 15 min at 12,000 rpm/min and 4 °C again. Finally, 20 μL of each sample was used for quality control analysis, and the remaining sample was used for LC-MS. When conducting chromatographic analysis, we placed the sample in an automatic sampler at 8 °C and passed it through an ultra-high-performance liquid chromatography system. The HSS T3 1.7 μm (2.1 × 50 mm) chromatographic column was used for gradient elution. The injection volume was 5 μL, the flow rate was 0.4 mL/min, and the column temperature was 40 °C. The chromatographic mobile phases were as follows: A: 5 mM ammonium formate in water; B: acetonitrile; C: 0.1% formic acid in water; and D: 0.1% formic acid in acetonitrile. The gradient elution procedure was as follows: 0–1 min, D changed linearly from 10% to 30%; 1–19 min, D changed linearly from 30% to 95%; 20–21 min, D changed linearly from 95% to 10%; and 21–25 min, D was maintained at 10%. A Thermo QEHF-X instrument (Waltham, MA, USA) was used for electrospray ionisation, and the cation–anion ionisation mode was used for mass spectrometry. The positive and negative ion spray voltage was 2.50 kV. The sheath gas was 50 arb, and the auxiliary gas was 13 arb. The capillary temperature was 325 °C, the capillary voltage was 35 V/−15 V, and the full scan was conducted at a resolution of 60,000 with a scope of 80–1000 *m*/*z*. CID was applied for secondary dissociation at a fragmentor voltage of 30 eV [39].

Metabolites were annotated according to Summer et al. (2007) [40].

### 4.3. Inhibition of F. asiaticum by B. subtilis S-16 and Its Fermentation Solution

The aseptic fermentation filtrate was obtained by filtering through a 0.22 μm aseptic filter. Then, 200 μL of the filtrate of each group was added to every Petri dish containing PDA. Subsequently, circular fungal blocks of CM-1 with a diameter of 8 mm were collected using a punch tool and placed at the plate centre for incubation at 25 °C for 5 days.

A total of 3 biological replicates were set up in this experiment, and 3 technical replicates were conducted. Meanwhile, the control group excluded aseptic fermentation filtrate. Finally, the colony diameter was measured by a ruler on the fifth day, and the inhibition rate was calculated as follows [41]:Inhibition rate (%) = (Control colony diameter − Treatment colony diameter)/Control colony diameter × 100%

Generally, an inhibition rate exceeding 50% indicates a strong inhibitory effect [9].

### 4.4. Effect of Seven Amino Acids on F. asiaticum CM-1 Growth

Seven secondary amino acid metabolites with antibacterial functions (Table 1) were purchased from Beijing Solaibao Technology Co., Ltd. (Beijing, China). They were configured into a 5 g/L solution and filtered with a 0.22 μm sterilised Millipore membrane to remove bacteria and obtained a sterile solution. CM-1 was placed in the centre of a PDA Petri dish and gently placed equidistant to the four sterilised Oxford cups on the four corners of the CM-1 plate. Then, 200 μL of each amino acid solution was injected into the Oxford cups before the pathogen cultivation. Subsequently, pathogens treated with 7 different amino acid solutions and without any treatment were incubated in a 28 °C microbial incubator at a constant temperature. The colony diameter was measured to establish the growth of CM-1 after 5 days, and then the inhibition rates were calculated. In this experiment, a total of 7 treatments were set up with 3 biological replicates per treatment and 3 Petri dishes per replicate.

The mycelia of the treatment group with the best inhibitory effect and the control group were selected with a sterilised inoculation needle and placed in a 2.5% glutaraldehyde fixing solution at 4 °C overnight for fixation. After fixation, samples were sent to the e-test service platform for scanning electron microscope imaging of the mycelia. The control group received no treatment.

### 4.5. Real-Time Quantitative Reverse Transcription Polymerase Chain Reaction

Real-time quantitative reverse transcription polymerase chain reactions (qRT-PCRs) were performed on *F. asiaticum* CM-1 to examine differences in the transcript levels of genes associated with root colonisation and strawberry resistance. Plant roots inoculated with NAD and CM-1 at the same time were selected at 0 and 72 hpi. The positive control group was treated with CM-1 alone, whereas the blank control group was treated with sterile water. Total RNA was isolated using an RNA extraction kit (TaKaRa Bio Inc., Kusatsu, Japan) according to the manufacturer’s protocol. First-strand cDNA was generated from RNA using Primer Script RT Master Mix (TaKaRa Bio Inc.). In the present study, the strawberry actin gene was used as an internal control, and all primers used for qRT-PCR are listed in Appendix A. qRT-PCR was carried out in a 10 µL reaction mixture containing 6 µL of TB Green ^®^ Premix Ex Taq™ (Tli RNaseH Plus; TaKaRa Bio Inc.), 0.4 µL of each primer, and 0.8 µL of templated DNA. All qRT-PCRs were performed using a QuantStudio5 real-time detection system (Thermo Fisher Scientific, Waltham, MA, USA). Each sample was analysed twice in three independent biological experiments. Relative expression levels of target genes were calculated according to the 2^−∆∆Ct^ method [42].

### 4.6. Control Effect of N-Acetyl-D-alloisoleucine on Strawberry Black Rot Disease

The spore concentration of the CM-1 isolate was adjusted to 1 × 10^6^ spores/mL and used to inoculate strawberry plants with five or six leaves planted in pots. The concentration of NAD was 5 g/L. In this experiment, three treatments and one control were set up in the experiment. Treatment I involved inoculating strawberry plants with NAD and *F. asiaticum* simultaneously; Treatment II involved inoculating strawberries with metalaxyl, reported to have good control effects on plant root rot disease [29], and *F. asiaticum* at the same time, and the concentration of metalaxyl was also 5 g/L. Treatment III involved inoculating strawberries with *F. asiaticum* alone. Each treatment consisted of 20 strawberry plants, with a total of 3 replicates set up. The blank control group was treated with sterile water. The dosage of NAD, the spore suspension of CM-1, metalaxyl, and sterile water was 20 mL each. Then, after 7 days, the disease index was calculated, and the control effect was assessed according to Abd-El-Kareem et al. [43].

### 4.7. Data Analysis and Plotting

SPSS software (version 25.0; IBM Corp., Armonk, NY, USA) was used for one-way analysis of variance (ANOVA), and then Duncan’s multiple range test was used after variance analysis. GraphPad software (version 8.0; Inc., San Diego, CA, USA) was used for mapping.

## 5. Conclusions

In recent years, China’s strawberry planting area and yield have continued to grow, making it one of the world’s largest strawberry-producing countries. In Inner Mongolia, the strawberry industry provides great economic benefits for agricultural facilities and occupies an important position in the region’s economy. The occurrence of a new disease, strawberry black root rot, seriously restricts the healthy development of the strawberry industry in Inner Mongolia. The continuous use of chemical agents not only pollutes the environment but also threatens food safety and human health. In this study, a secondary metabolite of *Bacillus subtilis*, which has inhibitory effects on *F. asiaticum* and can induce strawberry resistance to disease, was obtained. These results provide theoretical support for the research, development, and creation of new environmentally friendly biopesticides for the green prevention and control of plant diseases.

## Figures and Tables

**Figure 1 plants-14-00829-f001:**
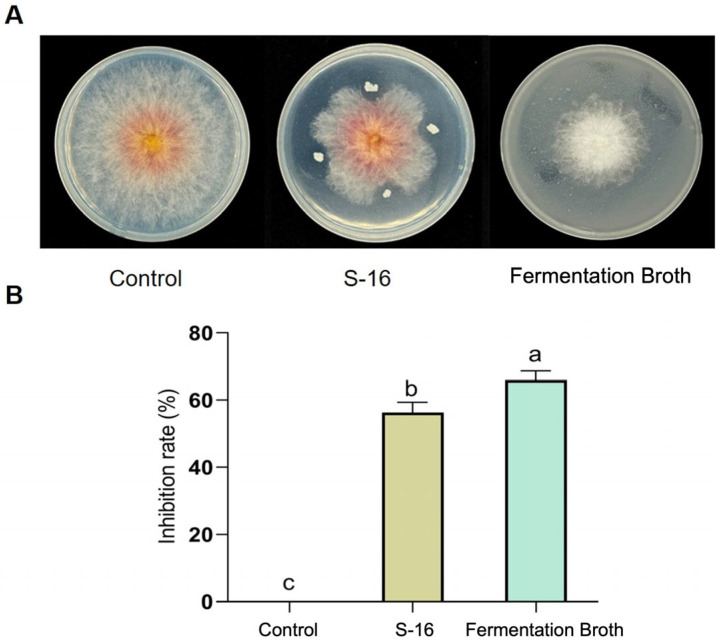
Inhibitory effects of *B. subtilis* S-16 and its fermentation broth on *F. asiaticum* CM-1. (**A**,**B**) Inhibitory effects and rates of *B. subtilis* S-16 and its fermentation broth according to plate experiments. Different letters on the bars (Duncan’s multiple range test, *p* < 0.05) represent significant differences.

**Figure 2 plants-14-00829-f002:**
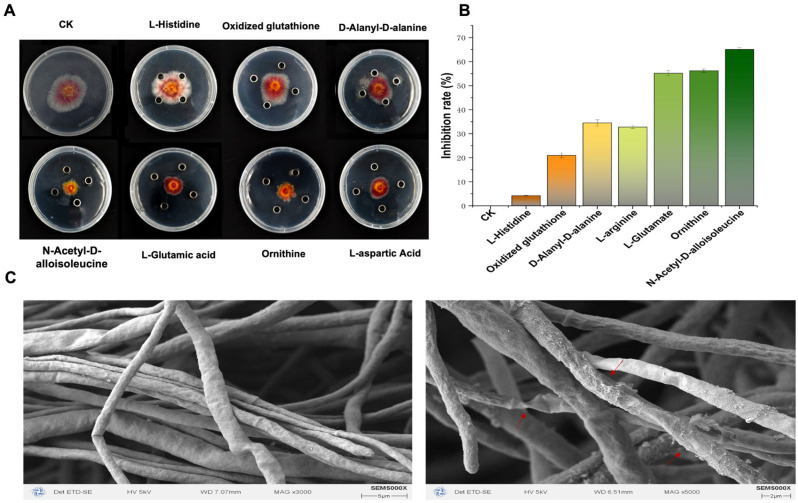
Inhibition of *F. asiaticum* by 7 amino acids and the morphology of mycelium treated with N-acetyl-D-alloisoleucine, photographed using an electron microscope. (**A**,**B**) Inhibition effects and rates of 7 amino acids; different letters on the bars (Duncan’s multiple range test, *p* < 0.05) indicate significant differences. (**C**) Mycelial morphology of *F. asiaticum* CM-1, photographed using electron microscopy. The treatment consisted of N-acetyl-D-alloisoleucine, while the control involved normal growth of *F. asiaticum* CM-1 without any treatment. The red arrows point to abnormal hypha. Significant differences are represented by different letters on the bars (Duncan’s multiple range test, *p* < 0.05).

**Figure 3 plants-14-00829-f003:**
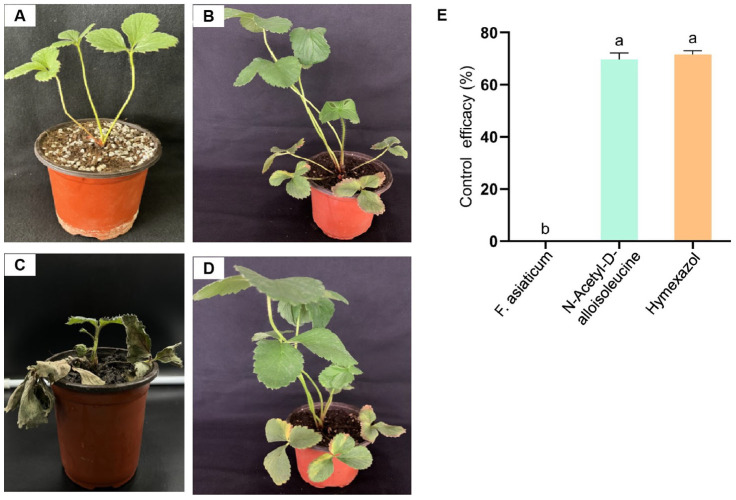
Disease control effect of N-acetyl-D-alloisoleucine on strawberry black rot. (**A**) Blank control. (**B**,**D**) The groups treated simultaneously with N-acetyl-D-alloisoleucine and Hymexazol together with *F. asiaticum*, respectively. (**C**) The plants inoculated with *F. asiaticum* as the positive control. (**E**) The control effect of N-acetyl-D-alloisoleucine and Hymexazol. Different letters on the bars (Duncan’s multiple range test, *p* < 0.05) represent significant differences.

**Figure 4 plants-14-00829-f004:**
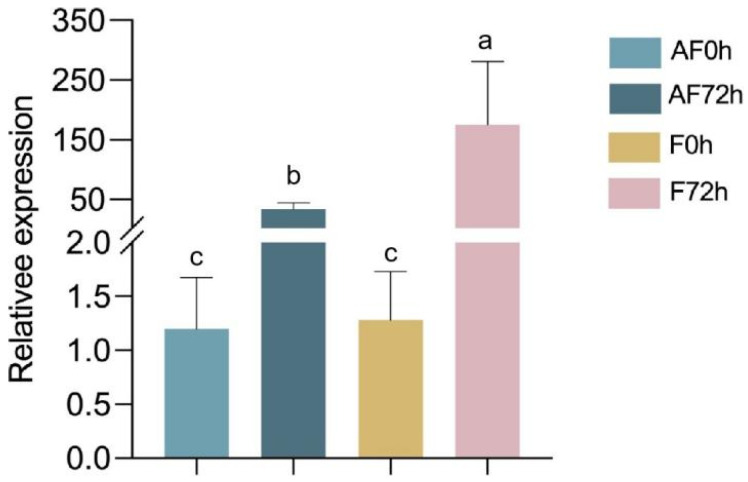
Relative expression levels of the gene EF-1α in strawberry roots at different time periods. AF0 and AF72 represent the plants concurrently inoculated with N-acetyl-D-alloisoleucine and spore suspensions of *F. asiaticum* assessed at 0 and 72 hpi, respectively. F0h and F72h represent the plants inoculated with *F. asiaticum* alone assessed at 0 hpi and 72 hpi, respectively. Different letters on the bars (Duncan’s multiple range test, *p* < 0.05) represent significant differences.

**Figure 5 plants-14-00829-f005:**
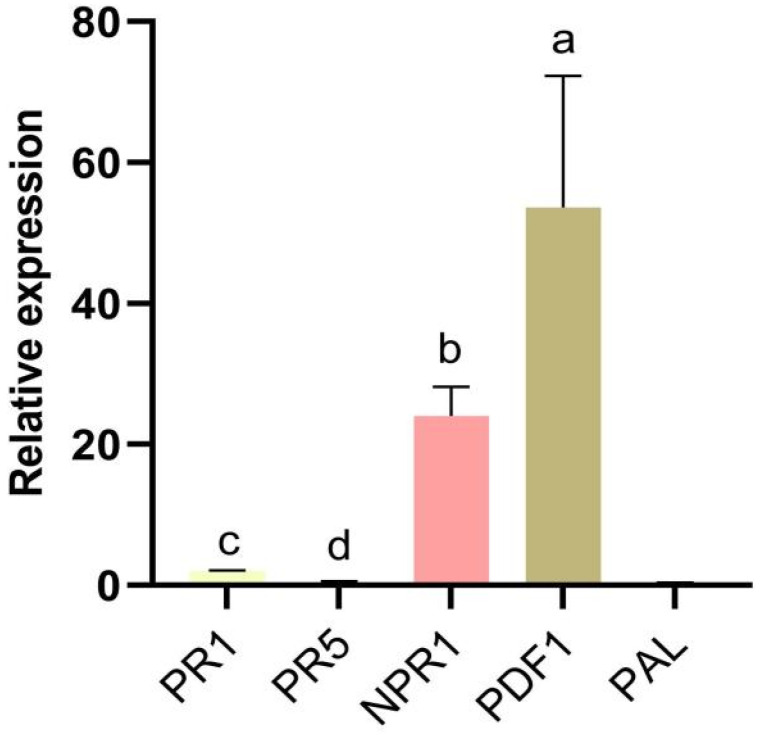
Relative expression of defence-related genes at 72 hpi after treatment with N-acetyl-D-alloisoleucine. Bars with the same lowercase letters are not significantly different according to the least significant difference multiple comparisons at *p* < 0.05.

**Figure 6 plants-14-00829-f006:**
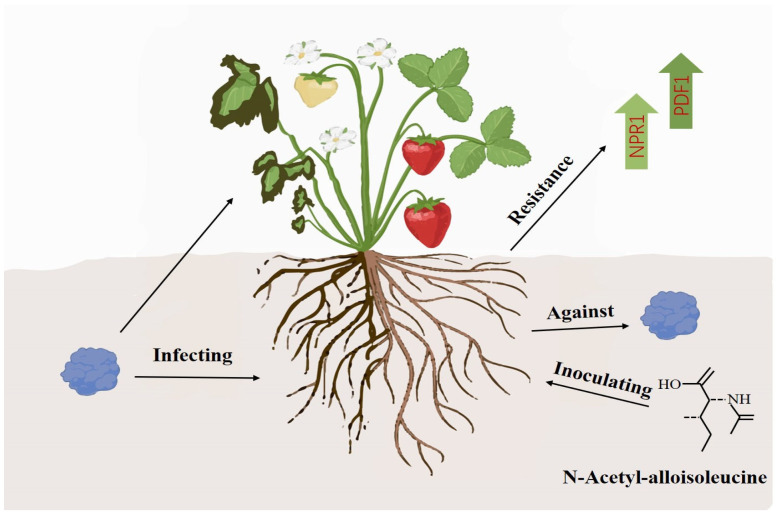
Model diagram of N-acetyl-D-alloisoleucine for controlling strawberry black root rot.

**Table 1 plants-14-00829-t001:** Screening of antimicrobial amino acids.

Compound	Regulation Quantity	VIP	*p*-Value	Reported Functions
Ornithine	down	1.46 × 10^−9^	0.0002	Promotes chlorophyll synthesis
L-histidine	down	1.28 × 10^−8^	0.0001	Promotes nitrogen metabolism in corn roots
N-acetyl-D-alloisoleucine	down	1.37 × 10^−9^	0.0013	Promotes corn root growth [3]
L-aspartic Acid	down	1.23 × 10^−9^	0.0002	Antibacterial anion
D-alanyl-D-alanine	down	1.45 × 10^−8^	0.0021	Antibacterial
Oxidised glutathione	down	1.28 × 10^−8^	0.0021	Steriliser
L-glutamic acid	down	1.71 × 10^−8^	0.0001	Signal conduction

## Data Availability

All additional datasets supporting the findings of this study are included within the article and Appendix A.

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
