# Peer review of "Mechanism of N-Acetyl-D-alloisoleucine in Controlling Strawberry Black Root Rot"

_plants, 2025, doi:10.3390/plants14050829_

Round 1
Reviewer 1 Report
Comments and Suggestions for Authors
In general, the manuscript "Mechanism of N-Acetyl-D-alloisoleucine in Controlling Straw-2 berry Black Root Rot" is well written and presents interesting results. I have some observations:
What I missed the most was how the 18 metabolites were annotated. Metabolites can be annotated at different levels (see Sumner et al., 2007 - doi:10.1007/s11306-007-0082-2), if commercial standards were used, or a database to compare the mass spectra. It is necessary include this information in the methodology. It would also be interesting to add the LC-MS/MS spectra of these metabolites to the supplementary material, comparing the m/z values ​​of the main peaks with theoretical peaks of databases.
What is "Oudemansia"? (line 31 and Figure 3). It is related to Oudemansiella species (Oudemansiella chenorum, Oudemansiella canarii, among others).
Minor corrections:
- Format "Candida albicans" in italic style (line 79);
- Specify "FB" in Figure 1;
- Specify "F", "AF" and "HM" in Figure 3;
- Change "LC/MS/MS" to "LC-MS/MS" (line 225);
- Add a space between "5" and "g/L" (lines 352 and 356);
- Change "After" to "after" (line 360).
- Standard the formatation of all references. Observe that the name of Journals are in italic style and abbreviated.
Author Response
Thank you for your reviewing manuscript. We have revised it according to your suggestions. Details are attached.

Reviewer 2 Report
Comments and Suggestions for Authors
The article provides information on the bioactivity of a bacterial isolate of Bacillus subtilis, and its metabolites, against the fungal pathogen Fusarium asiaticum, the agent of fusariosis in strawberry plants, in Inner Mongolia. Although the work has several points of interest, in some parts it is unclear due to incorrect linguistic expressions. It is recommended to have the entire article revised to improve the English language.
Comments on the Quality of English LanguageIt is recommended to review the entire article to improve its English language.
Author Response
Thanks for your valuable comments, we have improved the English language with the MDPI Author Service.

Reviewer 3 Report
Comments and Suggestions for Authors
The manuscript by Xu et al. investigates the effects of Bacillus subtilis S-16 and its fermentation solution on Fusarium asiaticum CM-1, the fungus causing black root rot in strawberries. The results show that both B. subtilis S-16 and its fermentation solution have a strong inhibitory capacity on the fungus, with up to 65.95% inhibition rates. The analysis of B. subtilis S-16 metabolites allowed the identification of 68 compounds, with amino acids being the most abundant. Of these, 7 were related to antibacterial properties. One of the most effective was N-Acetyl-D-alloisoleucine (NAD), which showed significant inhibition of F. asiaticum growth with a rate of up to 69.45%. NAD treatment also altered the morphology of the fungal mycelium, suggesting that NAD inhibits mycelium formation. In prevention experiments with strawberries, simultaneous treatment of roots with NAD and F. asiaticum spores showed similar effectiveness to the fungicide metalaxyl, with a controlled rate of 69.75%. In addition, NAD reduced the colonization of the fungus in strawberry roots and activated the expression of defense-related genes in plants, such as NPR1 and PDF1, indicating a strengthening of the plant's resistance against infection, demonstrating its potential as a controller for black root rot in strawberries, offering a sustainable and ecological alternative to traditional chemical methods. Therefore, the manuscript can be published, given the quality of its results. However, I would like to submit some suggestions and comments that the authors should review.
Although the text already provides a detailed description of many of the experimental methods, some aspects could still improve the study's clarity and reproducibility. Below, I mention the details that are missing, or that could be improved:
1. Regarding the experimental part, the pathogen F. asiaticum CM-1 is mentioned to have grown on PDA, but it would be helpful to specify the incubation conditions (temperature and exact duration of the culture). For the biocontrol strain B. subtilis S-16, it would be relevant to add details about the culture conditions before sowing in a liquid medium (for example, if an incubator with constant shaking is used, the transfer frequency, or the type of initial culture). Although it is mentioned that the PDA and LB medium are sterilized, it would be important to indicate how the sterility of the media is ensured before use (for example, autoclaving and filtering).
2. The LC-MS/MS analysis section mentions that the B. subtilis S-16 culture has grown to an OD600 of 1.0. It would be helpful to include an explanation of how the optical density is measured and how it is ensured that the culture is in the logarithmic phase of growth, as this may influence the metabolomics results. In addition, mass spectra are not presented in the supplementary information. If they have already been published, they should be cited in the references, and such references should be listed in Table S1.
3. Section 4.3 on the inhibition of F. asiaticum by B. subtilis S-16 mentions that the colony diameter is measured to calculate the inhibition rate. It would be important to indicate whether the measurement is performed on the same day or at regular intervals to check the progression of fungal growth. It might also be helpful to specify whether biological or technical replicates are used to ensure the reliability of the data.
4. In the F. asiaticum inhibition experiment by amino acids (section 4.4), I suggest specifying whether the amino acids were added before, during, or after pathogen inoculation to understand the interaction dynamics better. In addition, the method of measuring fungal growth in terms of biovolume or biomass is not mentioned, which could complement colony diameter measurements to provide more robust data.
5. Although the use of ANOVA and software for data analysis is mentioned, it would be important to indicate the type of post-hoc test used after variance analysis (e.g., Tukey, Bonferroni) to compare treatments.
Once these aspects have been reviewed, the manuscript can be considered for publication.
Author Response
Thanks for your hard work and valuable comments, we have made modifications according to your comments. The response letter is attached.
